# On-Site Soil Monitoring Using Photonics-Based Sensors and Historical Soil Spectral Libraries

Konstantinos Karyotis [1,2], Nikolaos L. Tsakiridis [1], Nikolaos Tziolas [3], Nikiforos Samarinas [1], Eleni Kalopesa [1], Periklis Chatzimisios [2,4] and George Zalidis [1,*]

1. Laboratory of Remote Sensing, Spectroscopy and GIS, School of Agriculture, Aristotle University of Thessaloniki, 57001 Thermi, Greece
2. Department of Information and Electronic Engineering, International Hellenic University of Greece (IHU), Alexander Campus, 57400 Sindos, Greece
3. Southwest Florida Research and Education Center, Department of Soil and Water Sciences, Institute of Food and Agricultural Sciences, University of Florida, 2685 State Rd 29N, Immokalee, FL 34142, USA
4. Department of Electrical and Computer Engineering, The University of New Mexico, Albuquerque, NM 87131, USA
* Correspondence: zalidis@auth.gr; Tel.: +30-2310-99-1779

**Abstract:** In-situ infrared soil spectroscopy is prone to the effects of ambient factors, such as moisture, shadows, or roughness, resulting in measurements of compromised quality, which is amplified when multiple sensors are used for data collection. Aiming to provide accurate estimations of common physicochemical soil properties, such as soil organic carbon (SOC), texture, pH, and calcium carbonates based on in-situ reflectance captured by a set of low-cost spectrometers operating at the shortwave infrared region, we developed an AI-based spectral transfer function that maps fields to laboratory spectra. Three test sites in Cyprus, Lithuania, and Greece were used to evaluate the proposed methodology, while the dataset was harmonized and augmented by GEO-Cradle regional soil spectral library (SSL). The developed dataset was used to calibrate and validate machine learning models, with the attained predictive performance shown to be promising for directly estimating soil properties in-situ, even with sensors with reduced spectral range. Aiming to set a baseline scenario, we completed the exact same modeling experiment under laboratory conditions and performed a one-to-one comparison between field and laboratory modelling accuracy metrics. SOC and pH presented an $R^2$ of 0.43 and 0.32 when modeling the in-situ data compared to 0.63 and 0.41 of the laboratory case, respectively, while clay demonstrated the highest accuracy with an $R^2$ value of 0.87 in-situ and 0.90 in the laboratory. Calcium carbonates were also attempted to be modeled at the studied spectral region, with the expected accuracy loss from the laboratory to the in-situ to be observable ($R^2 = 0.89$ for the laboratory and 0.67 for the in-situ) but the reduced dataset variability combined with the calcium carbonate characteristics that are spectrally active in the region outside the spectral range of the used in-situ sensor, induced low RPIQ values (less than 0.50), signifying the importance of the suitable sensor selection.

**Keywords:** MEMS; Vis-NIR; harmonization; transfer learning; light-based technologies; soil health; EU soil mission; onsite digital tools; exploratory modelling; carbon farming; VNIR; SWIR

## 1. Introduction

Over the last several years, the European Union has worked diligently towards farming modernization, with a key objective of mitigating soil degradation due to climate change and poor farming practices. Farmers, aligning themselves with this trend, employ contemporary approaches that simplify and optimize the farming procedure, and seek cost-effective solutions for each stage of the farming chain, with a goal to significantly reduce time and costs [1].

Infrared spectroscopy is a branch of physics and specifically of wave optics that is related to the study of the structure and composition, as well as the nature of the spectra of matter and radiation, covering the visible (Vis), near infrared (NIR), and shortwave infrared (SWIR) parts of the electromagnetic spectrum. It is the science that studies light as a function of wavelength, as the result of reflections due to molecular oscillations. Soil spectra can be used to determine various soil properties [2], such as the nature and content of organic matter or soil organic carbon (SOC), as well as the calcium carbonate the soil is likely to contain, soil moisture content, texture, soil acidity, and nutrient content. The properties that can be determined via spectroscopy are dependent on the specific instrumentation and measurement method used, since each part of the electromagnetic spectrum studied, provides information for different soil parameters. Vis–NIR–SWIR regions can be used for the qualitative and quantitative analysis of soils [3], and present significant advantages for providing estimations immediately without the need for time-consuming pre-treatments and chemical reagents [4].

In this context, soil spectroscopy has already been considered as an important support to traditional laboratory procedures that can facilitate the development of large soil datasets, and due to their non-destructive nature can mitigate environmental impact, with satisfactory measuring accuracy [5]. With the advent of photonics-based solutions, such as micro-electro-mechanical systems (MEMS), soil monitoring can be performed on a much finer time scale, quantifying the effect of applied practices on the soil ecosystem [6–8]. In particular, soil spectroscopy leveraged with participatory sensing techniques can provide a cost and time-effective solution for the accumulation of information related to critical agro–environmental-related parameters, from parcel-level to large-scale areas. Despite the progress in artificial intelligence (AI) algorithms and the development of novel sensing systems, the adoption of portable sensors is still not optimal, and, to our knowledge, there are no prior attempts to retrieve valuable insight from the utilization of portable spectrometers by different users under real field conditions.

With soil spectroscopy, high amounts of data, in terms of points or images, can be processed with the use of models from which conclusions are drawn. Various statistical and machine learning (ML) techniques, such as partial least-squares regression (PLSR), random forests (RF), the Cubist algorithm, or support vector regression models (SVRs) have been evaluated and proven to be effective in linking the shape of the spectral curve to a targeted soil characteristic [9,10]. Conventional ML models, or with even higher accuracy, the newly developed deep learning techniques radically transform the soil spectral regression by better serving feature learning, and thus produce accurate estimations, even simultaneously as multi-output models [11]. Furthermore, AI models have also been utilized in recent studies [12] to develop transfer functions in order to predict the field from laboratory spectral signatures in a way to match the two spectral datasets and eliminate the effects of ambient factors in the field. Since AI can be a suitable modeling tool, especially for cases where the number of explanatory variables substantially exceeds the number of observations, statistical modeling is considered as a reliable scientific approach and has been applied to various studies for spectra treatment. Ackerson et al. [13] employed an external parameter orthogonalization algorithm to minimize the effect of soil moisture on the in-situ spectra for clay content prediction, while the application of the same technique along with piecewise direct standardization, direct standardization, and generalized least squares weighting, were used by Yang et al. [14] to eliminate the external effect on the spectra for SOC modelling purposes. The presented results are promising, but until today, the studies that have been conducted are covering limited soil types and further investigation is needed, as stated in [15], where a set of mathematical techniques for eliminating the effects of moisture content from soil spectra signatures have been reviewed.

With initiatives, such as LUCAS, GEO-Cradle, or other open soil spectral libraries (SSLs), soil can be studied as a data-driven approach [16,17] by identifying solid statistical relationships between laboratory spectral signatures and soil parameters. Augmenting regional SSLs through spiking is a common technique, but when it comes to merging in-situ

and laboratory SSLs, especially without the application of any means of standardization, burdens, such as the different usage protocols or the effects of environmental factors arise, resulting in low model performance [18,19]. This further limits the exploitation of the existing open SSLs from spectrometers that were designed to operate in situ, since most of them have been developed under laboratory conditions. To overcome this obstacle, various efforts have been made in the past to develop a universally accepted protocol for SSL development. Currently, the "IEEE P4005 standards and protocols for soil spectroscopy" is expected to support such an effort, as a result of joined forces from more than 100 soil spectroscopy experts, but still, the cases of laboratory and in-situ soil spectroscopy are accurately discriminated as two distinct set-ups.

Our main objective is to leverage in-situ soil spectroscopy with the elimination of the ambient factors effect through the application of AI-based functions, linking the laboratory with the field-collected data. More concretely, through the exploration of different ML modeling techniques, we exploit existing high-quality and expensive SSLs to enhance the predictive performance of low-cost in-situ sensors. To ensure that the quality of on-site collected data is not compromised, we developed an AI-driven outlier detection mechanism and adopted the spectral standardization methodology based on internal soil standard (ISS) usage [20]. In the next sections, we provide a detailed description of our effort to merge in-situ soil spectral signatures with existing laboratory SSLs, providing accurate AI-based estimations of key physical and chemical soil properties, as these were tested with data collected during three field excursions for soil sampling and data acquisition in areas with different soil characteristics from different countries. The results of the two modeling iterations (laboratory spectra and in-situ spectra) per soil property were compared to evaluate the potential of the low-cost portable SWIR spectrometer to provide accurate estimations of each soil property.

## 2. Materials and Methods

### 2.1. Overall Methodology

A set of 280 topsoil samples were collected by 20 soil surveyors, distributed in three countries (Cyprus, Lithuania, and Greece), chemically analyzed in the laboratory, and scanned in-situ with a portable low-cost SWIR spectrometer (Nirone S2.2 @1750–2150 nm, Spectral Engines GmbH, Steinbach, Germany—Fabry-Perot interferometer with a spectral resolution of 20 nm), and at the lab with an analytical benchtop spectrometer (PSR + 3500 @350–2500 nm, Spectral Evolution Inc., Lawrence, MA, USA—512 element silicon photo-diode array (350–1000 nm) with a spectral resolution of 2.8 nm, two 256 element InGaAs photodiode arrays: (970–1910 nm) with a spectral resolution of 8 nm and (1900–2500 nm) with a spectral resolution of 6 nm). Prior to the field visit, a pre-calibrated SWIR spectrometer was delivered to each soil surveyor to perform the in-situ measurements. The pre-calibration step using the ISS ensures the inter-standardization of the devices between themselves, as well as with other SSLs. The SSL developed, meaning that the spectral measurements combined with the chemical analysis were used to calibrate the ML regression algorithms for the estimation of the targeted physical and chemical properties, with the sensors' bands representing the topsoil spectrum to be used as explanatory variables. To increase the variability of both spectral and physicochemical measurements, we augmented the compiled in-situ SSL with the existing open SSL of GEO-Cradle [21]. The GEO-Cradle regional SSL contains Vis-NIR-SWIR spectra and physicochemical analyses from nine countries around the Balkans, eastern Mediterranean, and North Africa. For the purposes of our work, only the Greek and Cypriot samples were used. Since this SSL consists exclusively of laboratory spectra, we developed an AI-based transformation that maps the in-situ spectra captured by the portable SWIR spectrometer, to the corresponding spectral domain of the spectral signatures captured by the analytical benchtop spectrometer in the laboratory. This technique's objective was dual; at first to eliminate the effect of ambient factors on the spectral response, and secondly to harmonize the in situ and laboratory datasets, after the correction was applied to the spectrum through the application of ISS standardization. In

the meantime, for each spectral measurement collected in-situ, a quality assessment was performed with a developed classifier that was used as an outlier detection mechanism. The resulting dataset was used to estimate the values of a set of physical and chemical soil properties based on AI techniques. Figure 1 demonstrates an overview of the workflow followed to produce the final estimation of the targeted soil properties.

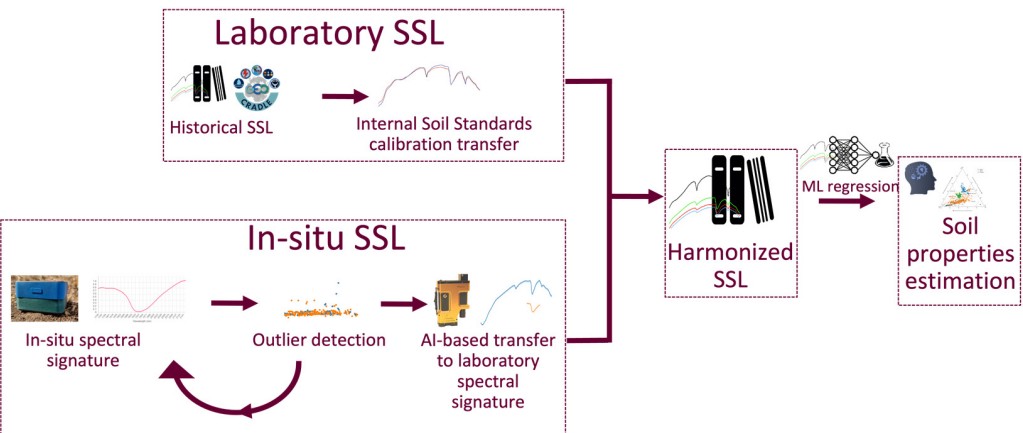

**Figure 1.** Methodological flow for the soil properties estimation based on in-situ spectral signatures.

## 2.2. Data Collection

An extended data collection excursion took place during the summer of 2021, in three pilot regions of Cyprus, three in Lithuania, and one in Greece (Figure 2). Their total extent is about 25 km² and 280 points were distributed according to the conditional Latin hypercube selection (cLHS) [22] algorithm which was applied as follows: A three-year-long Sentinel-2 time-series was shaped for each pilot area, containing pixels corresponding to cultivated land that was bare [23] at the time of the image capture, i.e., no vegetation or other plant materials are present. The followed approach is described by Castaldi et al. [24] and according to whom the pixels of cloud-free images were masked based on spectral indices of the normalized difference vegetation index (NDVI) at less than 0.25, the normalized burn ratio 2 (NBR2) at less than 0.075, and B3 > B2 and B2 > B1 (where B represents the Sentinel-2 band). The Sentinel-2 multi-temporal values of each bare soil instance were converted to a bare soil composite by calculating the median value of each pixel. The feature space of the resulting dataset was expanded with ancillary information containing topographical and physicochemical properties of open source datasets which are a NASA SRTM digital elevation model [25] and ISRIC soil grids SOC content and soil texture [26]. Then, the cLHS algorithm was applied to define the pixels corresponding to locations that will be used for the field visits from where the topsoil spectra were collected with the use of the portable spectrometer, while an amount of topsoil was collected for physical and chemical analyses.

## 2.3. Chemical Analyses

For the sampled soil, in addition to the spectral analyses, chemical analyses were performed as well, and more concretely, 40 topsoil (0–20 cm) physical samples were collected from each region, amounting to 280 samples, that were analyzed for the determination of the following physical and chemical properties:

- Texture (Sand, Silt, and Clay—%);
- Soil organic carbon content (SOC—%);
- pH;
- Calcium carbonates (CaCO$_3$—g/kg).

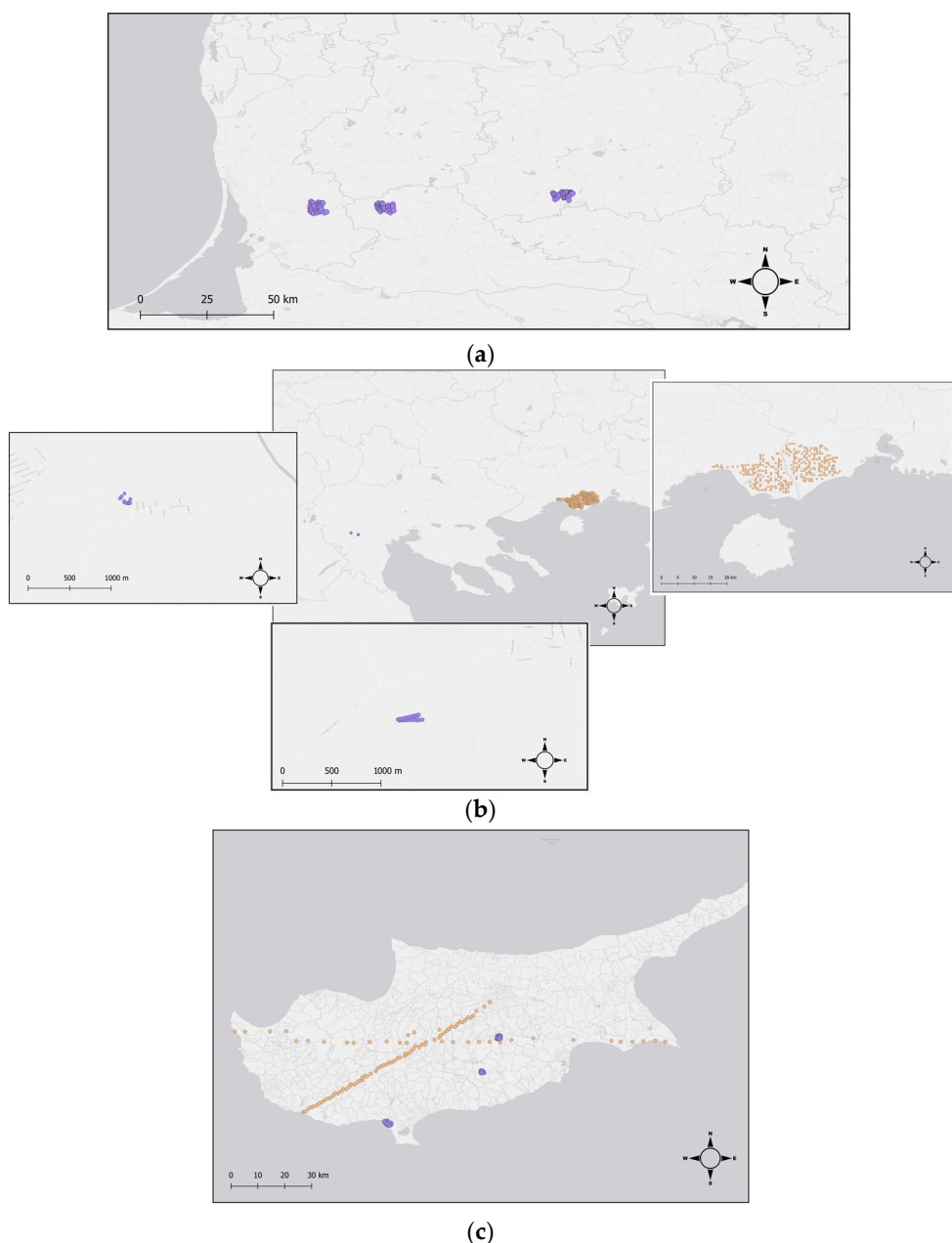

**Figure 2.** Spatial distribution of the sampling locations (purple) and GEO-Cradle samples (yellow) in (**a**) Lithuania, (**b**) Greece, and (**c**) Cyprus.

Soil texture was determined using the Bouyoucos hydrometer method [27], the SOC content was measured with the Walkley–Black method [28], and soil acidity was measured as pH, the values of which were determined through the saturated paste extract method [29], while the concentration of $CaCO_3$ was measured with the use of a calcimeter.

### 2.4. Spectral Analyses

#### 2.4.1. Spectral Measurements

The samples were transferred to the laboratory and were pretreated, meaning that they were air-dried naturally and then passed through a 2 mm sieve. Then, they were placed inside a completely opaque "dark box" with two 75 W tungsten lamps, and were measured at laboratory conditions with the analytical benchtop spectrometer which operates at the spectral region of 350 to 2500 nm. To ensure that the produced SSL is reusable and

compatible with existing open SSLs, the measuring protocol proposed by Ben-Dor et al. [30] was used, which is also in accordance with the protocol that the GEO-Cradle regional SSL developed. More specifically, standardization was applied with the use of two ISSs, Wiley Bay (WB) and Lucky Bay (LB), adopting the idea from Pimstein et al. [20], suggesting that to minimize systematic deviations between laboratories, the notion of internal standards of wet chemistry can be transferred to soils.

All samples were measured in-situ with the portable spectrometers, and the above-described measuring scheme was employed to the extent possible, and more specifically, before delivering the spectrometers to the operators, the ISSs were measured with each device, and the correction factors were calculated, aiming to reduce between-device measuring deviations. The topsoil reflectance was measured by directly placing the sensor facing the ground. Prior to measuring, stones, roots, litter, or other non-soil materials were removed. The devices used were Spectral Engines Nirone S2.2, which were modified according to the suggestions and insights provided by Karyotis et al. [31], since this study demonstrated that the capacity of the sensor to reliably capture soil spectral signatures can be significantly increased if the field of view is adjusted accordingly. To ensure the proper operation under field conditions (e.g., waterproof, dust resistant, etc.), we adjusted the sensing device by creating an extension component using 3D printing technology and attached it to the self-sensor as shown in Figure 3.

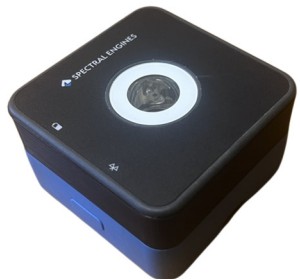

**Figure 3.** Customized sensor: A 3D printed bracket was adjusted to the sensor to increase the field-of-view.

### 2.4.2. Outlier Detection

The in-situ spectra measurements were collected by the different operators bearing different devices and under different environmental conditions. To ensure that the collected data were of high quality, even though explicit equipment usage instructions were given to each user, we developed an AI classifier for the outlier detection. With this tool, we aimed to verify that each spectral signature was collected properly, meaning that the equipment was used according to the predefined usage protocol, and no hardware failure occurred. Furthermore, a photo was captured from each exact scanned location, which, along with the corresponding Exif metadata, was checked before accepting any spectral signature as valid.

To achieve this, we identified that for soil spectroscopy, measurements with an outlying behavior were all non-soil materials that a user might try to scan, or measurements that were taken without conforming to the usage protocol. To this end, a collection of about 200 spectral signatures from non-soil objects was created (Figure 4) and merged with soil measurements captured with the portable spectrometer, and various supervised and unsupervised classifiers were evaluated for their potential to separate soils from non-soils. A set of classifiers, including the K-nearest neighbors classifier, support vector machines, Gaussian process classifier, decision tree, and random forest, were evaluated using accuracy, precision, and recall as evaluation metrics, and as a result, the selected classifier was the ensemble method based on the abovementioned algorithms, that was used for testing each spectral signature before any particular analysis or modeling was performed. A recent review by Ganaie et al. (2022) [32] summarizes the techniques for ensemble learning that combine several individual models to obtain a better predictive performance.

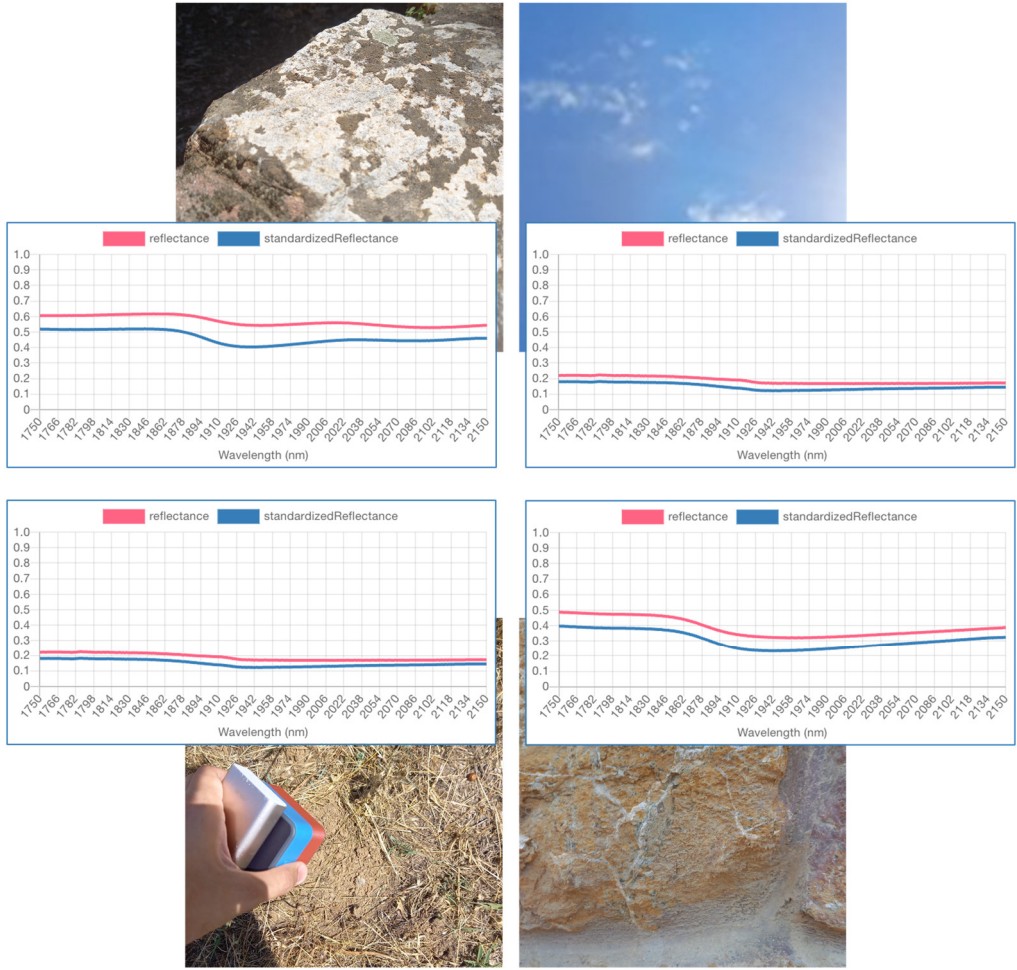

**Figure 4.** Instances of the collection of non-soil objects for the outlier dataset creation and their reflectance spectra before and after standardization.

### 2.4.3. Ambient Factors Effect Elimination

Soil moisture, shadows, the existence of rocks, or other non-soil materials, such as vegetation residues or any kind of litter affect the soil's spectral signature. To eliminate the effect of these factors, a second AI-based component was developed, which can be considered as a signal denoising function applied to the in-situ spectrum, exploiting the spectral relationships between the spectra from the lab and in-situ. This mechanism can be briefly described as a one-dimensional convolutional neural network (CNN) that targets the elimination of the effects passed to the spectral signature shape and height by ambient factors (such as moisture or shadows). The function maps the array of spectra collected in-situ (exploratory variables) to the array of spectra captured in laboratory conditions (response variables). We randomly split the dataset at a 70–30 ratio and the network's hyperparameters were tuned after a grid search, and the best combination was selected based on the optimal value of RMSE, calculated over the two categories of spectra. More concretely, different activation functions were tested, including relu, sigmoid, softplus, softsign, hyperbolic tangent, selu, and elu, along with different batch and kernel sizes ranging from 3 to 5 and from 13 to 17, respectively. The resulting SSL containing the transformations of the in-situ spectra to the laboratory spectra can be merged with any historical SSL (such as GEO-Cradle that was used in our experiment) given that it was measured according to the same protocol as the one we used for our laboratory analysis.

### 2.5. Modeling Soil Properties

The effects of soil characteristics on specific spectral domains can be represented as a vector composed of thousands of reflectance measurements. More specifically, local extrema of the spectral signature correspond to the fundamental frequencies and overtones of the soil components, which are caused by the increase of their molecules' kinetic energy that absorb the transmitted electromagnetic radiation. To quantify each soil characteristic, a linkage with the shape of the spectral signature must be established. Machine learning regression algorithms were proven to be reliable solutions for this mission, and if fine-tuned and trained over a dataset representing well the variability of the soil properties to be modeled, high estimation accuracy can be achieved. For our case, we tested the random forest [33] regression algorithm, the Cubist algorithm [34,35], and the SVR algorithm [36], as provided by the scikit-learn Python library and the Cubist R wrapper.

To highlight the important parts of the spectrum, a set of pretreatment techniques were applied for the scatter correction and spectral derivation functions, which was mainly the calculation of absorbance spectra as the decimal logarithm of inverted reflectance. Furthermore, the Savitzky–Golay [37] smoothing and differentiation filter was applied to the initial spectral curve and their first or second-order derivatives, fitting a third polynomial with the least squares technique to a window length of 101 for the laboratory spectra and 25 for the field spectra, respectively, expressed in nanometers, to a set of point observations, thus reducing the noise through data smoothing, but also keeping the spectral tendency without inducing any distortions. A detailed overview of those spectral pre-treatments may be found in [38].

We evaluated the three mentioned algorithms and all possible combinations of spectral preprocessing for each soil property and selected the modeling scenario that produced the optimal values of standard performance metrics. Prior to proceeding to any model fitting, a representative subset equal to 30% of the observations was selected and left out through the Kennard–Stone algorithm based on in-situ spectral similarities [39], and was used as an independent set for assessing the accuracy of every developed model, which was quantified with standard model assessment metrics, such as the coefficient of determination ($R^2$), the root mean square error (RMSE), and the ratio performance of the interquartile range (RPIQ). For the same train-test split, the same spectral pre-processing and modelling procedure was applied to the laboratory spectra, which serve as an estimation performance upper boundary due to the analytical nature of the spectral signature capture in laboratory conditions.

## 3. Results and Discussion

### 3.1. Chemical Results

A large range of variability was captured over the set of monitored parameters, and in particular over the soil texture, since the soils were sampled from different places with different soil properties. As shown in Figure 5, the three groups of soils can be easily distinguished. Here, soils originating from Cyprus and Lithuania are predominantly Leptosols and Regosols [40,41], with medium to high clay content. The samples originated from Cyprus are labeled according to the United States Department of Agriculture (USDA) classification system as clay loam, silty clay loam, or silty loam, and the ones from Lithuania are mainly classified as sandy clay loam, loam, or clay loam. Greek soils are mostly Fluvisols and Entisols [42], and are classified mainly as loamy and sandy loamy, based on the soil texture, which mostly lies to the fact that the GEO-Cradle soil samples, which are the majority of the dataset instances, are distributed mainly around the Nestos river estuaries, a sandy area with low clay content.

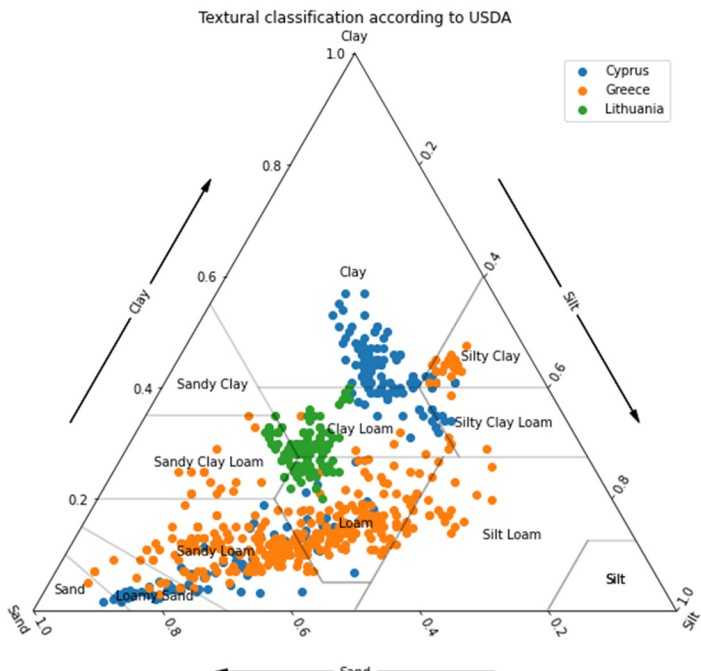

**Figure 5.** Soil sample's textural classification according to the USDA.

The descriptive statistics of the soil data are presented in Table 1. The average pH value (7.87) is slightly higher than expected for croplands, which typically ranges from 5.5 to 7.5 [43], and this can be explained due to the alkaline soils of one of the pilot areas in Cyprus (around Lefkara, the Larnaka region), and the combination of high concentrations of $CaCO_3$, high clay fraction, and low SOC content [44], the pH value in this region exceeds 8.4. Added to that, Lithuanian soils from the test sites were either neutral or marginally alkaline. Furthermore, GEO-Cradle does not provide pH measurements for Greek soils, thus the high pH values for the Cypriot soil affect the calculated average value of the SSL. SOC presents a low average value of 0.98%, and when combined with the high clay content that prevails in most parts of the studied regions, it signifies a high SOC sequestration potential that is not currently achieved [45–47]. As for the correlations (Figure 6), positive correlations have been calculated as Pearson's coefficient between clay and all other properties, with the strongest observed between the pH and $CaCO_3$. SOC is not directly correlated with pH or $CaCO_3$, while carbonates and acidity are positively and strongly correlated.

### 3.2. Spectral Measurements

ISS measurements from different devices demonstrated a low variance in the captured reflectance, as shown in Figure 7a, and in combination with the absence of strong absorption bands, hence no artifacts were introduced to the spectral signature to be corrected, proving the suitability of the selected materials as soil standards. This technique was considered necessary since 20 different devices were used for the accumulation of in-situ spectral measurements, but it must be noted that operator-related systematic effects were not eliminated, since the ISSs were measured at the laboratory. The application of the standardization decreased the spectral differences between the reflectance captured with the handheld and the benchtop (reference) spectrometers by an order of magnitude. As differences were measured in terms of RMSE, the unstandardized error compared to the reference values ranged from 11% to 18% while the error that corresponded to the standardized reflectance measurements was reduced significantly, ranging from 1% to 4% for the different spectral ranges (Figure 7b).

**Table 1.** Descriptive statistics of the chemical analyses for the augmented dataset, GEO-Cradle, and the collected samples.

| Soil Attribute | Sand (%) | Clay (%) | Silt (%) | pH | CaCO$_3$ (g/kg) | SOC (%) |
|---|---|---|---|---|---|---|
| Augmented dataset (GEO-Cradle and collected samples) | | | | | | |
| Min | 9 | 1.5 | 6 | 5.95 | 0 | 0 |
| 1st quartile | 31 | 11 | 26 | 7.67 | 0 | 0.63 |
| Median | 45 | 18 | 31 | 7.95 | 10 | 0.91 |
| Mean | 45.87 | 22.24 | 31.89 | 7.87 | 91.32 | 0.98 |
| 3rd quartile | 59 | 32 | 39 | 8.09 | 80 | 1.32 |
| Max | 89 | 57 | 62 | 10.07 | 815 | 3.8 |
| Standard Deviation | 17.85 | 13.48 | 10.04 | 0.43 | 165.18 | 0.59 |
| Collected samples | | | | | | |
| Min | 12 | 10 | 13 | 7.04 | 3.00 | 0.70 |
| 1st quartile | 24 | 28.25 | 26 | 7.70 | 12.00 | 0.95 |
| Median | 32 | 35 | 29 | 7.95 | 36.50 | 1.29 |
| Mean | 34.68 | 34.87 | 30.46 | 7.86 | 142.46 | 1.35 |
| 3rd quartile | 45 | 43 | 34 | 8.07 | 280.00 | 1.59 |
| Max | 75 | 57 | 48 | 8.27 | 590.00 | 3.80 |
| Standard Deviation | 14.43 | 10.12 | 7.12 | 0.28 | 168.16 | 0.49 |
| GEO-Cradle | | | | | | |
| Min | 9 | 1.50 | 6 | 5.95 | 0.00 | 0.00 |
| 1st quartile | 43 | 9 | 24 | 7.46 | 0.00 | 0.47 |
| Median | 54 | 13 | 33 | 7.97 | 2.00 | 0.70 |
| Mean | 53.66 | 13.47 | 32.89 | 7.91 | 55.98 | 0.74 |
| 3rd quartile | 63.80 | 17 | 41 | 8.35 | 20.00 | 0.99 |
| Max | 89 | 48 | 62 | 10.07 | 815.00 | 3.66 |
| Standard Deviation | 15.74 | 7.02 | 11.54 | 0.72 | 153.44 | 0.54 |

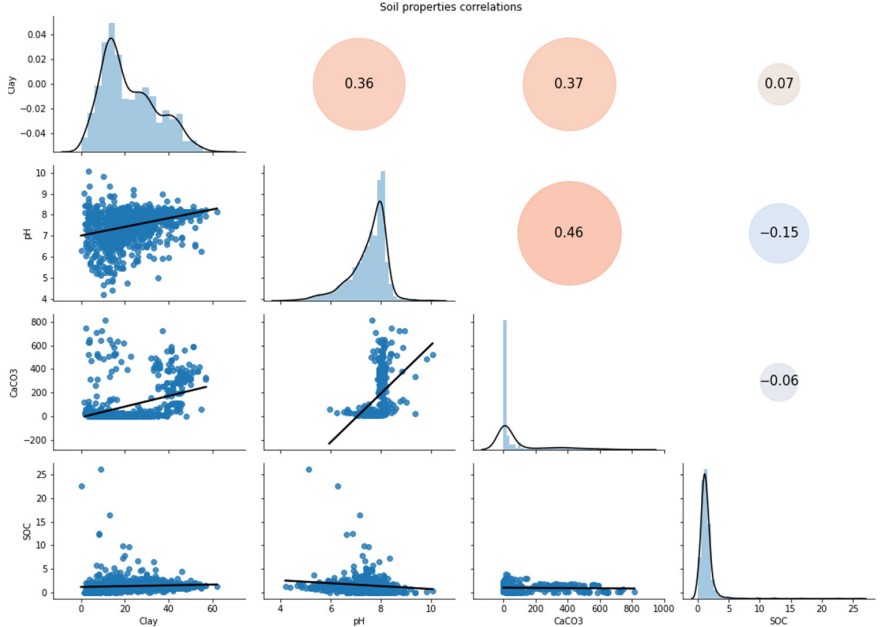

**Figure 6.** Probability density estimations and pairwise scatterplots and correlations of the soil properties as analyzed in the laboratory.

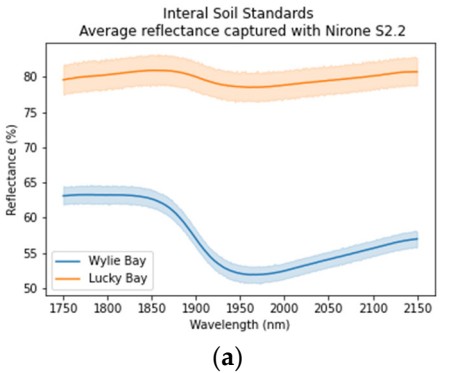 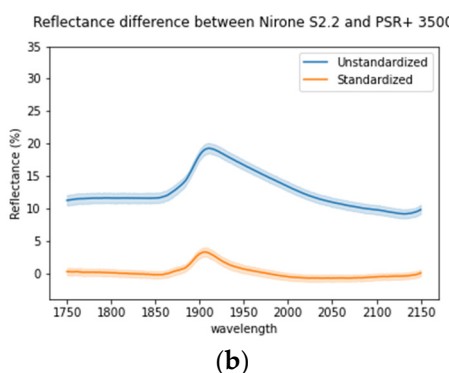

(**a**) (**b**)

**Figure 7.** (**a**) Wiley Bay and Lucky Bay spectral signatures (average reflectance and 95% confidence interval), as captured on Nirone S2.2 (**b**) Spectral differences between the reflectance captured with PSR + 3500 and Nirone S2.2 before (Unstandardized) and after standardization (Standardized).

Spectral measurements from the collected soil samples present high spectral variability, including soils covering both low and high albedo, compiling a diverse spectral dataset. This diversity can be depicted both from the laboratory and in-situ spectral curves, as shown in Figure 8. Furthermore, it can be highlighted that the in-situ measurements have lower reflectance than the laboratory measurements. This mainly has to do with the existence of soil moisture, shadows, roughness, or other factors, the effect of which were eliminated with the soil pretreatment applied before the laboratory spectral measurements.

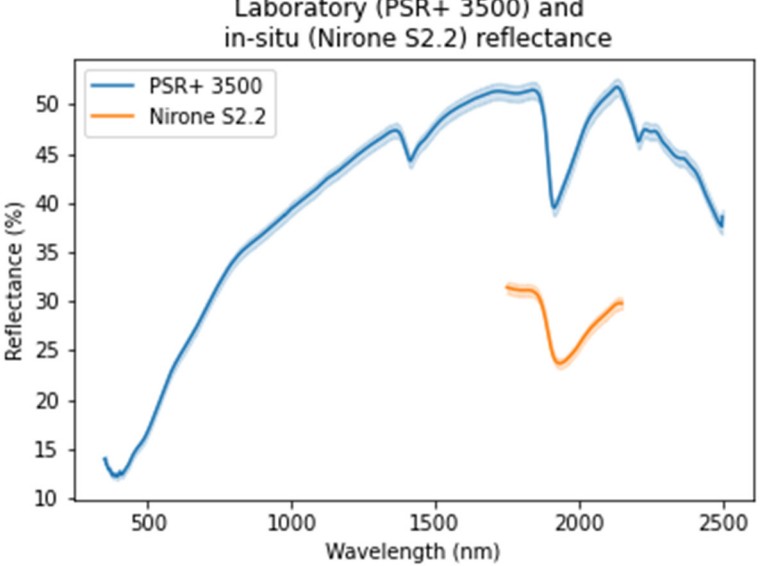

**Figure 8.** The reflectance of the Vis-NIR laboratory analysis, and the SWIR in situ analysis after the standardization with the ISS was applied (average reflectance and 95% confidence interval).

### 3.3. Outlier Detection

Standard classifying accuracy metrics (precision, accuracy, and recall) were used to assess each classifier that was developed for the detection of the spectral signatures with outlying behavior. The ensemble method combining all of the classifiers outperformed each other's modeling attempt, providing an accuracy of 94.48%, a precision of 95.28%, and a recall score of 97.58%. As shown in the corresponding confusion matrix of Table 2, only nine instances were erroneously classified (six false positives and three false negatives), while the classification outcomes from the ensemble method are presented in Figure 9.

**Table 2.** Confusion matrix of ensemble method for identifying measurements with outlying behavior.

|  |  | Actual Value | |
|---|---|---|---|
|  |  | **True** | **False** |
| Predicted value | True | 33 | 6 |
|  | False | 3 | 121 |

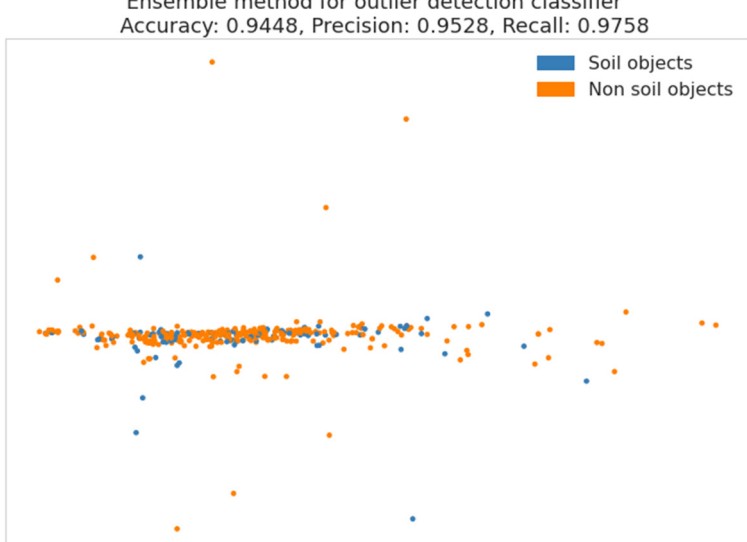

**Figure 9.** Ensemble method classifier for the outlier detection predictions. Dimension reduced with a principal components analysis (PC1 and PC2).

As many field surveyors were engaged in the field campaign, a few instances of device misuse or malfunction were detected by the outlier detection mechanism, resulting in the need for measurement repetition. This signifies the necessity for real-time inspection and conformity mechanisms when it comes to field campaigns that are carried out by medium to large-sized groups, and further enables users with no prior explicit experience to participate in such crowdsourcing missions and vastly increase the scale of studied regions.

### 3.4. Ambient Factors Effect Elimination

A dense grid search approach was followed for hyperparameter tuning of the CNN-based spectral transfer function from in-situ to the laboratory. The median RMSE presents an increasing trend when the batch size increases for any given activation function (Figure 10a), whereas activation function selection influences the interquartile range of estimated RMSE, resulting in deviations with high variability from reference values for different batch size combinations. Furthermore, kernels of different sizes affect the RMSE (Figure 10b), and thus the kernel and batch sizes of 15 and 5, respectively, and the scaled exponential linear unit as activation function was selected, resulting in the development of a CNN architecture that will minimize the spectral distance between the in situ and laboratory reference spectral curves.

The correction shown in Figure 11 has a considerable effect on minimizing the differences between the laboratory and in situ spectral signatures. This effect is mostly present around the high absorption band of 1920 nm, with the smallest effect visible at the two limits of the handheld spectrometer's spectral range due to the lack of numerous surrounding bands. The resulting spectral signatures created a regional SSL, which could subsequently be extended through spiking local samples from open SSLs that have been developed with the same measuring protocol, such as GEO-Cradle.

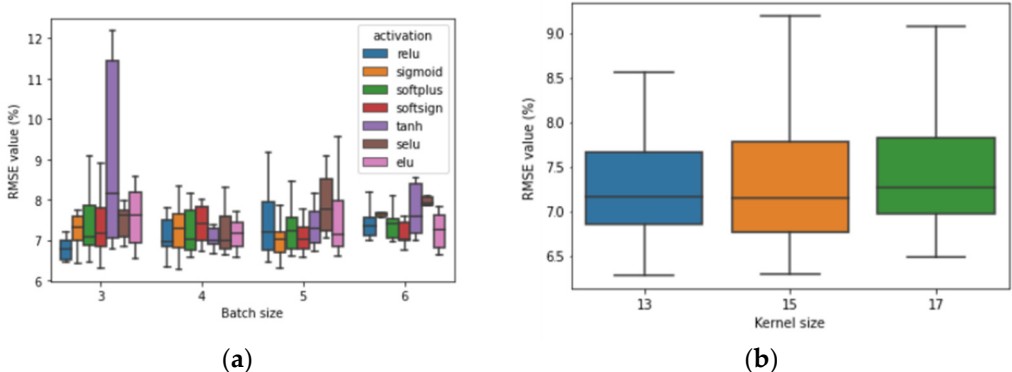

(**a**)  (**b**)

**Figure 10.** Boxplots of the RMSE values over (**a**) different batch sizes per activation function, and (**b**) different kernel sizes.

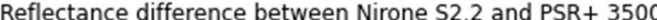

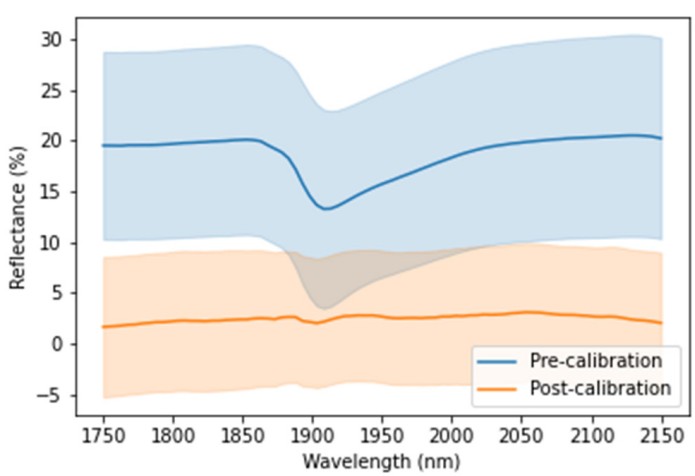

**Figure 11.** Difference between the in situ spectra and their transformation to laboratory reference spectral values (average reflectance and 95% confidence interval).

*3.5. Modeling Assessment*

We split the dataset with the Kennard–Stone algorithm to a 70-30 ratio, aiming to achieve a representative train-test split. This can be verified through Figure 12, where the groupings of each region per soil property provide similar distributions (probability density estimations) in the sense that both the train and the test subset present high frequencies at similar positions of the property values. As previously discussed, the high pH values observed in Cyprus are mainly a result of high $CaCO_3$ values, leading to different distributions of pH and $CaCO_3$ between Cyprus and the other two cases of Greece and Lithuania, and more specifically to great differences in high values and the value range of these two properties.

Clay content was estimated with a very high accuracy, with the metric values presenting a slight decrease when the modeling field data ($R^2$ = 0.87, RMSE = 4.13%, RPIQ = 3.86) were compared to the laboratory data ($R^2$ = 0.90, RMSE = 3.66%, RPIQ = 4.03). The best fitting model for the laboratory case was the Cubist algorithm after applying SNV normalization to the reflectance spectra, while for the in-situ case, the SVR algorithm was chosen with the Savitzky–Golay (1st order derivative) filtering. Many clay minerals show intense absorption in the SWIR region, resulting from vibrations of hydroxyl groups and water molecules [48], and thus sensors operating at SWIR are suitable for their detection and their content determination.

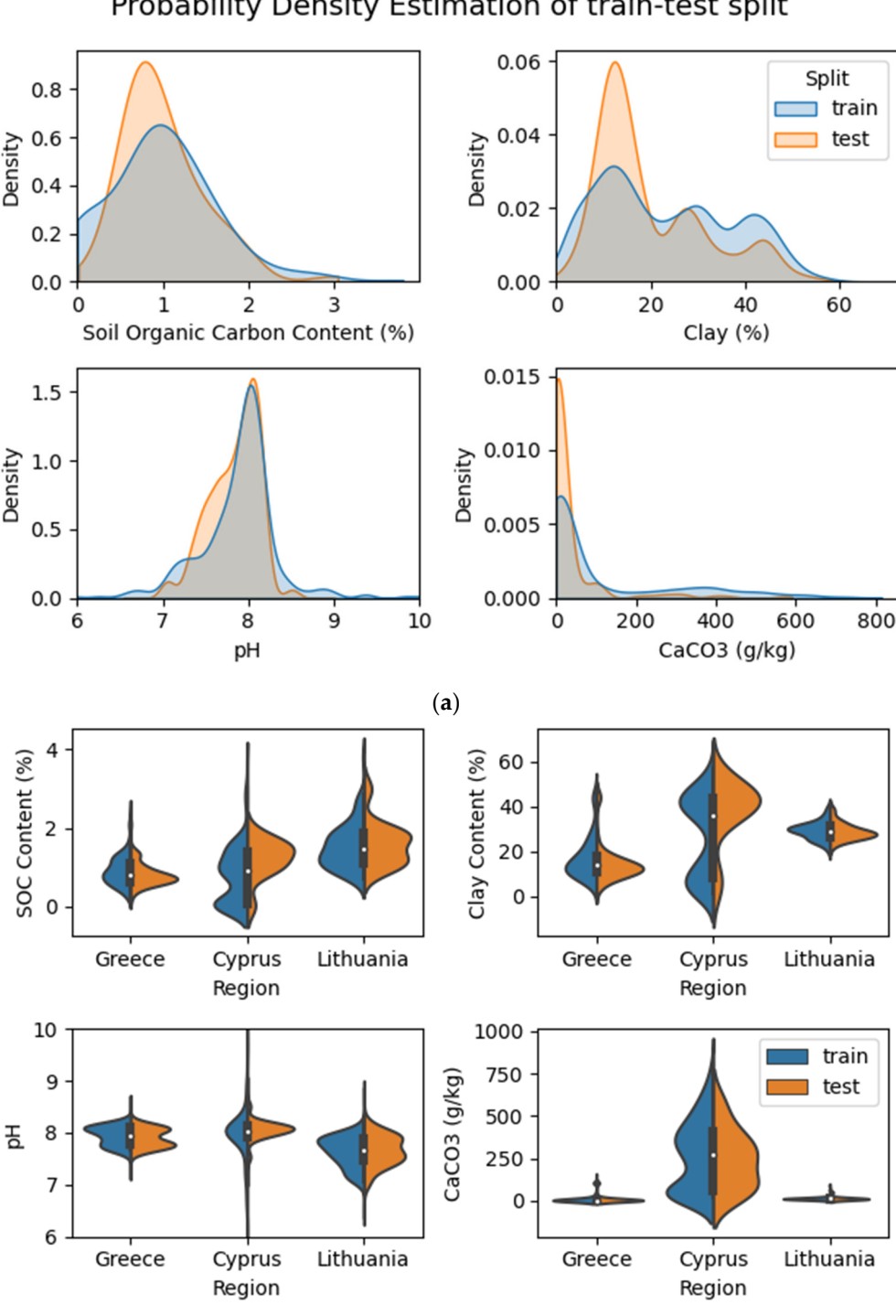

**Figure 12.** Probability density estimation and the box plot of the train-test split (**a**) for the whole dataset and (**b**) per region for each soil property.

For the estimation of the SOC content, the fitted models were based on the random forest regression which was applied to scatter the corrected reflectance spectra with the Savitzky–Golay (1st order derivative) and normalized with SNV. Here, the metrics value decrease from the lab ($R^2$ = 0.63, RMSE = 0.37%, RPIQ = 1.81) to the field ($R^2$ = 0.43, RMSE = 0.36%, RPIQ = 1.48) is significant since the bands correlated with SOC are fewer in the spectral range covered by the Nirone S2.2 sensor. The important spectral bands for the

SOC content estimation are mostly located around the visible range (400–700 nm), with two bands (1752 nm and 2056 nm) falling inside the in-situ sensor's spectral region, according to Stenberg et al. [49].

The results of the pH and calcium carbonates do not allow for drawing safe conclusions since the laboratory modeling unveiled correlations between the spectrum and the targeted property ($R^2 = 0.41$ for pH and $R^2 = 0.89$ for $CaCO_3$), but the induced bias was high. From Figure 13, we can observe that the trend was identified with in-situ modelling, corresponding to low and high reference values, respectively (with $R^2 = 0.32$ for pH and $R^2 = 0.67$ for $CaCO_3$), but with a relatively high bias and a few outliers observed for both soil parameters. Other studies claim that the pH and $CaCO_3$ can be efficiently modeled by Vis-NIR-SWIR spectra [42,50], but for our case, it is very likely that high values of RMSE produced by our modeling are mostly related to special characteristics of the dataset used. The experiment mostly contains soil samples with a neutral pH, except for one region with very high concentrations of $CaCO_3$ that leads to soil alkalinity, thus the distribution of the pH and $CaCO_3$ declines from the normal or exponential distribution and presents a fat tail to the high values. Overtones of fundamental absorbance bands of $CaCO_3$ can be found within the spectral range of field spectrometers, as the 7th (at 2012 nm), 6th (at 1909 nm), and 4th (at 1796 nm) overtones of 710 cm$^{-1}$, 873 cm$^{-1}$, and 1392 cm$^{-1}$, respectively [51], justifying the identified correlations of the collected spectrum with the carbonates, and thus, the cross-correlations with the pH. Performance metrics are presented in Table 3.

**Figure 13.** Scatterplots of the observed predicted values of the soil properties for the laboratory (**upper**) and in-situ (**lower**) modelling cases (best fitting line and 95% confidence interval).

**Table 3.** Accuracy metrics for the best fitting model per property and experimental set-up.

| Set-Up | Selected Model | Property | $R^2$ | RMSE | RPIQ |
|--------|----------------|----------|-------|------|------|
| Laboratory | Cubist | Clay | 0.90 | 3.66% | 4.03 |
| | RF | SOC | 0.63 | 0.29% | 1.81 |
| | SVR | pH | 0.41 | 0.22 | 1.91 |
| | Cubist | $CaCO_3$ | 0.89 | 30.63 (g/kg) | 0.46 |
| In-situ | SVR | Clay | 0.87 | 4.13% | 3.86 |
| | RF | SOC | 0.43 | 0.36% | 1.48 |
| | RF | pH | 0.32 | 0.25 | 1.87 |
| | Cubist | $CaCO_3$ | 0.67 | 54.08 (g/kg) | 0.18 |

As expected, accuracy loss was observed in the modelling in-situ spectra compared to the pretreated laboratory analysis, and that is mostly related to (i) the limited spectral

range of the in-situ sensor compared to the analytical instrument used in the laboratory, and (ii) the effect of ambient factors, which was partially eliminated. We demonstrated that the clay content estimation from the in-situ spectra is feasible with values of $R^2 = 0.87$ and RPIQ = 3.86, and our findings are in accordance with Rossel et al. [52] where strong correlations of Vis-NIR spectra were found both in the laboratory and in-situ conditions. According to Ji et.al. [53], the SOC can be accurately estimated based on the in-situ Vis-NIR spectra with $R^2$ greater than 0.75 signifying the importance of NIR region at less than 1750 nm, which was not covered in the presented work. Compared to other studies that report a value loss of the $R^2$ metric from 0.66 to 0.39 in modeling the SOC from the laboratory to the in-situ [54], our results slightly outperformed this experiment which is mostly explained by the attempt to minimize the effect of ambient factors, in combination with the SOC variability that our dataset captured. A MEMS based system (Neospectra Scanner @ 1350–2500 nm) was also assessed at a recent work [7] for its predictive capacity of the same set of soil properties as our study, but only in laboratory conditions, demonstrating that MEMS can be used effectively to substitute expensive laboratory equipment, while in Shen et al. [8], the MEMS sensor used in the present work was also evaluated and successfully used for the estimation of a wider set of soil properties and health indicators at laboratory conditions. The estimation of the pH on site was slightly outperformed from the laboratory, with an observable decrease in $R^2$ value of 0.9 and a very small increase in the RMSE value of 0.03 and a small loss in RPIQ of 0.04. The accuracy metrics achieved are in accordance with the previous work [55], where an RMSE of 0.23 and RPIQ of 2.01 are reported for modeling the in-situ NIR spectra.

The results of the modeling attempt of $CaCO_3$ are weaker compared to the aforementioned soil properties and to research efforts [56] that outperform the one presented in this paper. This is mostly attributed to the dataset characteristics, that although it contains soils from different regions and types, the special characteristics of alkaline Lefkara-Cyprus soils have a strongly asymmetrical distribution, since the other samples have very low $CaCO_3$ values. This distribution's particularity can be considered as a "real world's" attribute rather than a special case, since the same can be observed also in the pan-European soil survey of LUCAS where the $CaCO_3$ distribution was positively skewed, creating a long tail to the right part of the curve corresponding to the higher values [16].

In general, our methodology achieved comparable modeling results between the laboratory and in-situ, with the latter slightly underperforming the former, and this agrees with the previous study by Priori et al. [7], where a standard full Vis-NIR-SWIR spectrometer was compared with a portable MEMS sensor, covering the range from 1350–2500 nm, and attaining a better predictive performance for all soil features. Moreover, the RPIQ values of clay, SOC, and pH present a relative value loss of less than 19% (which corresponds to the value loss of the SOC modelling), in exchange of a significantly decreased number of explanatory variables, which in our case are the spectral bands captured from each sensor (reducing the 2150 bands of the analytical spectrometer used at the laboratory to the 400 bands of the MEMS sensor used in-situ). The results presented can also be attributed to the application of the AI-based technique that was applied for the ambient factor removal, for which the selected kernel size effectively captured the intercorrelations of the consecutive bands, enabling the model to preserve the interclass distance of the spectra. Different modeling architectures, in terms of algorithms (such as auto-encoders) or used metrics as loss-functions (such as the Mahalanobis distance) can be tested to further enhance the elimination of the effects applied to the spectrum during in-situ measuring.

## 4. Conclusions

The presented work provides insight into the potential to estimate various soil properties based on the accurate measurement of in-situ topsoil reflectance with the use of low-cost sensors capturing very few spectral bands, narrow spectral ranges, or with low spectral resolution. We demonstrated the far-reaching opportunities using effective light-based technologies for the in-situ analysis and AI to quantify four physio-chemical properties in

soils. Our work paves the way for a new wave of promising research to tackle unprecedented challenges in onsite and near-real time estimation of soil parameters. Deep learning models were employed to bridge the gap between laboratory and in-situ spectroscopy, and as a result past efforts of developing SSLs based on laboratory measurements can now be further exploited to support the calibration and validation of ML models that accurately assess topsoil properties. Still, the importance of developing a globally approved sensing protocol is emphasized.

We performed a one-to-one comparison between the laboratory and in situ spectral signatures with measurements obtained with a full range, high-resolution, analytical spectroradiometer (Spectral Evolution, PSR + 3500) and a narrow-ranged handheld spectrometer (Spectral Engines, Nirone S2.2). The applied transformation induced a correction to the spectra acquired in situ that resulted in a significant spectral difference drop, which can be reflected as a reduction of the mean absolute error from 20% to 2.5%, which was calculated over an independent set. This technique minimizes the effects of ambient factors on the spectral signatures and helps the development of a new dataset of "transformed" spectra, which can be used to extract useful information about the soil's characteristics.

The use of spectrometers with a limited spectral range compared to analytical devices mainly used in laboratories can provide accurate estimations of soil properties, such as clay, SOC, or pH, that attain an RPIQ > 1.4 in all cases, while $CaCO_3$ needs to be further explored. Furthermore, the comparison of modeling results between the benchtop spectrometer and the handheld device showed a small decrease in accuracy when using the latter, which signifies that the tested soil properties can be accurately estimated with the selection of the appropriate device, in terms of spectral range. As the proposed architecture is modular, it may be tailored to various environmental observations or soil spectrometers, and as a result to radically reform environmental monitoring methodologies.

**Author Contributions:** Conceptualization, K.K., N.L.T. and E.K.; methodology, K.K., N.L.T. and N.T.; software, N.L.T.; validation, K.K., N.S. and N.T.; formal analysis, N.S., P.C. and G.Z.; investigation, E.K., K.K., N.L.T. and N.T.; resources, E.K. and G.Z.; data curation, K.K. and N.L.T.; writing—original draft preparation, K.K.; writing—review and editing, N.L.T.; visualization, N.L.T.; supervision, E.K., P.C. and G.Z.; project administration, E.K. and G.Z.; funding acquisition, E.K. and G.Z. All authors have read and agreed to the published version of the manuscript.

**Funding:** The research leading to these results has received funding from the European Community's Framework Programme Horizon 2020 under grant agreement No 870378, project DIONE.

**Data Availability Statement:** The data presented in this study are available upon request from the corresponding author (G.Z.).

**Acknowledgments:** The authors would also like to thank the National Paying Agency of Lithuania and the Cyprus Agricultural Payments Organization in assisting the field survey for data collection.

**Conflicts of Interest:** The authors declare no conflict of interest. The funders had no role in the design of the study; in the collection, analyses, or interpretation of data; in the writing of the manuscript; or in the decision to publish the results.

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
