# Peer review of "On-Site Soil Monitoring Using Photonics-Based Sensors and Historical Soil Spectral Libraries"

_remotesensing, doi:10.3390/rs15061624_

Round 1

Reviewer 1 Report

This manuscript presents an interesting work on on-site soil information prediction using low-cost spectrometers together with laboratory-based soil spectral library. Authors found positive results for clay prediction while the performance for SOC, pH and CaCO3 was still limited, calling further work on suitable sensor selection. This manuscript is generally easy to follow while it still has a great room to be improved: (1) there are many typos should be corrected; (2) the knowledge gap should be better summarized; (3) the methodology is not well present; (4) the discussion needs to be further improved. Therefore, a major revision is suggested.

Specific comments are listed below:

Line 31: R2 should be corrected as R2. Please correct same issue in other relevant positions.

Line 33: How weak it was? Please use number to support your statement.

Line 38: I can not find references in the main text. Please insert them in the right positions.

Line 63: Please remove ,, here.

Line 66: large-scale, not large-scaled.

Lines 85-89: This is not the only available solution. One can use EPO, DS or PDS to remove the effect of external effect on the in-situ spectra, thus export laboratory spectra to be used by integrating laboratory-based soil spectral library. Some examples are listed below:

Ackerson, J.P., Demattê, J.A. and Morgan, C.L., 2015. Predicting clay content on field-moist intact tropical soils using a dried, ground VisNIR library with external parameter orthogonalization. Geoderma, 259, pp.196-204.

Yang, M., Chen, S., Li, H., Zhao, X. and Shi, Z., 2021. Effectiveness of different approaches for in situ measurements of organic carbon using visible and near infrared spectrometry in the Poyang Lake basin area. Land Degradation & Development, 32(3), pp.1301-1311.

Lines 116-118: Spectral resolution should be mentioned for two spectrometers.

Lines 148-149: Which data were used to conduct cLHS algorithm for selecting sampling locations?

Lines 223-241: Please more details about the CNN model to eliminate ambient factors effect. If I understand correctly, you only have one exploratory variable for each band. So how can CNN perform well with only one variable?

Lines 267: You have two spectrometers and two different spectral measurement conditions, so which spectra were used to conduct Kennard-Stone algorithm? If you perform Kennard-Stone algorithm for each kind of spectra, then the validation set will change which will make the model evaluation not comparable.

Figure 6: Please remove _H20 after pH. In addition, % can also be removed after SOC.

Figure 12: We observed a great difference between train and test sets. Do you think it will introduce a large uncertainty to the model evaluation?

Figure 13: This figure has a poor resolution, please present in better quality.

Lines 406-421: There is not deep discussion in this manuscript. Please strengthen this part regarding to current limitations and perspectives.

Author Response

The authors would like to thank reviewer #1 for his/her feedback provided. We strongly regard that the reviewer's suggestions are very accurate and helped increase the quality of our manuscript.  Considering the reviewer’s suggestion for a major revision we reformatted the paper by:

Updating the manuscript with more recent references, including reviewer’s suggestions (lines XX – XX – XX).

Finally, addressing all the points(e.g. typos, references etc.)that have been raised in the reviewers’ comments and we performed a proof-reading before the second submission. 

Please see the attached for our detailed answer

Reviewer 2 Report

This manuscript entitled "On-site soil Monitoring Using Photonics-Based Sensors and Historical Soil Spectral Libraries" by Konstantinos Karyotis et al. attempted to use the AI model to transfer the on-site measured soil spectra to indoor spectra, and the regression methods such as RF were used to establish models for estimating soil components such as SOC and PH. In general, the model is effective, but there are some problems in the manuscript, which are described as follows. Some points need to be clarified, and some statements need further modification. Suggestions for revision are given below.

1.     No reference marks throughout the manuscript, which must be reinserted by the author.

2.     What is the spectral resolution and number of bands of Nirone S2.2 versus PSR+3500? What method of resampling is used?

3.     Is SSL only used for Outlier detection? How can you assure that the SSL-built model can distinguish anomalous spectra more effectively, given that the SSL library use a spectrometer that is very different from your present portable instrument?

4.     Line 217, what is the meaning of “ensemble method”? Is it evaluated based on the precision of KNN, RF, etc.?

5.     Is CNN network a simple LeNet? How to divide the training set and test set?

6.     Is RMSE or MSE as the only loss function to determine the accuracy of the model prone to overfitting problems? The smaller the RMSE, the better the generated spectrum will be? By employing the CNN generating model, have the original spectral features been removed? For instance, the actual interclass distance of interior spectra is considerable, whereas the interclass distance of spectra generated on-site is small? In other words, will the spectral discrepancies caused by soil composition variances also be eliminated?

7.     Information on the inversion accuracy of the training set and test set for different inversion methods should be presented in a table.

8.     What is the inversion accuracy of the raw spectrum(uncorrected spectrum)?

9.     Are feature selection methods applied before inversion model building because of the high redundancy of hyperspectral data?

e.g

Li, H., Liang, Y., Xu, Q., Cao, D., 2009. Key wavelengths screening using competitive adaptive reweighted sampling method for multivariate calibration. Analytica Chimica Acta 648, 77-84.

Ou, D., Tan, K., Lai, J., Jia, X., Wang, X., Chen, Y., Li, J., 2021. Semi-supervised DNN regression on airborne hyperspectral imagery for improved spatial soil properties prediction. Geoderma 385, 114875.

10.  The figure needs to be replaced with a clear one, such as Figure 13.

11.  The discussion section is missing.

12.  The conclusion section should be shortened.

Author Response

The authors would like to thank the reviewer for his/her feedback in order to improve the clarity of our manuscript. At the attached file we present an itemized point-by-point reply to all comments and suggestions raised by the #2 reviewer. 

Round 2

Reviewer 1 Report

The revised version has been improved a lot and the reply to my comments are ok. Therefore, I suggest that it can be accepted for publication.

Author Response

Dear reviewer,

thank you very much for your effort and the provided comments.

Best regards,

George

Author Response

(The authors gave the same response as above.)
